# ON THE DYNAMICS OF TRAINING ATTENTION MODELS

**Haoye Lu, Yongyi Mao & Amiya Nayak**
School of Electrical Engineering and Computer Science
University of Ottawa
Ottawa, K1N 6N5, Canada
`{hlu044, ymao, nayak}@uottawa.ca`

## ABSTRACT

The attention mechanism has been widely used in deep neural networks as a model component. By now, it has become a critical building block in many state-of-the-art natural language models. Despite its great success established empirically, the working mechanism of attention has not been investigated at a sufficient theoretical depth to date. In this paper, we set up a simple text classification task and study the dynamics of training a simple attention-based classification model using gradient descent. In this setting, we show that, for the discriminative words that the model should attend to, a persisting identity exists relating its embedding and the inner product of its key and the query. This allows us to prove that training must converge to attending to the discriminative words when the attention output is classified by a linear classifier. Experiments are performed, which validate our theoretical analysis and provide further insights.

## 1 INTRODUCTION

Attention-based neural networks have been broadly adopted in many natural language models for machine translation (Bahdanau et al., 2014; Luong et al., 2015), sentiment classification (Wang et al., 2016), image caption generation (Xu et al., 2015), and the unsupervised representation learning (Devlin et al., 2019), etc. Particularly in the powerful transformers (Vaswani et al., 2017), attention is its key ingredient.

Despite its great successes established empirically, the working mechanism of attention has not been well understood (see Section 2). This paper sets up a simple text classification task and considers a basic neural network model with the most straightforward attention mechanism. We study the model's training trajectory to understand why attention can attend to the discriminative words (referred to as the topic words). More specifically, in this task, each sentence is treated as a bag of words, and its class label, or topic, is indicated by a topic word. The model we consider involves a basic attention mechanism, which creates weighting factors to combine the word embedding vectors into a "context vector"; the context vector is then passed to a classifier.

In this setting, we prove a closed-form relationship between the topic word embedding norm and the inner product of its key and the query, referred to as the "score", during gradient-descent training. It is particularly remarkable that this relationship holds irrespective of the classifier architecture or configuration. This relationship suggests the existence of a "synergy" in the amplification of the topic word score and its word embedding; that is, the growths of the two quantities promote each other. This, in turn, allows the topic word embedding to stand out rapidly in the context vector during training. Moreover, when the model takes a fixed linear classifier, this relationship allows rigorous proofs of this "mutual promotion" phenomenon and the convergence of training to the topic words.

Our theoretical results and their implications are corroborated by experiments performed on a synthetic dataset and real-world datasets. Additional insights are also obtained from these experiments. For example, low-capacity classifiers tend to give stronger training signals to the attention module. The "mutual promotion" effect implied by the discovered relationship can also exhibit itself as "mutual suppression" in the early training phase. Furthermore, in the real-world datasets, where perfect

delimitation of topic and non-topic words does not exist, interesting training dynamics is observed. Due to length constraints, all proofs are presented in Appendix.

## 2 RELATED WORKS

Since 2019, a series of works have been published to understand the working and behaviour of attention. One focus of these works pertains to understanding whether an attention mechanism can provide meaningful explanations (Michel et al., 2019; Voita et al., 2019; Jain & Wallace, 2019; Wiegreffe & Pinter, 2019; Serrano & Smith, 2020; Vashishth et al., 2020). Most of these works are empirical in nature, for example, by analyzing the behaviours of a well-trained attention-based model (Clark et al., 2019), or observing the impact of altering the output weights of the attention module or pruning a few heads (Michel et al., 2019; Voita et al., 2019), or a combination of them (Jain & Wallace, 2019; Vashishth et al., 2020). Apart from acquiring insights from experiments, Brunner et al. (2019) and Hahn (2020) show theoretically that the self-attention blocks lacks identifiability, where multiple weight configurations may give equally good end predictions. The non-uniqueness of the attention weights therefore makes the architecture lack interpretability.

As a fully connected neural network with infinite width can be seen as a Gaussian process (Lee et al., 2018), a few works apply this perspective to understanding attention with infinite number of heads and infinite width of the network layers (Yang, 2019; Hron et al., 2020). In this paper, we restrict our study to the more realist non-asymptotic regime.

## 3 PROBLEM SETUP

**Learning Task** To obtain insights into the training dynamics of attention models, we set up a simple topic classification task. Each input sentence contains $m$ non-topic words and one topic word indicating its topic. Note that a topic may have multiple topic words, but a sentence is assumed to include only one of them. Assume that there are $J$ topics that correspond to the mutually exclusive topic word sets $\mathbb{T}_1, \mathbb{T}_2, \cdots, \mathbb{T}_J$. Let $\mathbb{T} = \bigcup_{j=1}^{J} \mathbb{T}_j$ be the set of all topic words. The non-topic words are drawn from a dictionary $\Theta$, which are assumed not to contain any topic word.

The training set $\Psi$ consists of sentence-topic pairs, where each pair $(\chi, y)$ is generated by (1) randomly pick a topic $y \in \{1, 2, \cdots, J\}$ (2) pick a topic word from set $\mathbb{T}_y$ and combine it with $m$ words drawn uniformly at random from $\Theta$ to generate the sentence (or the bag of words) $\chi$. In this task, one aims to develop a classifier from the training set that predicts the topic $y$ for a random sentence $\chi$ generated in this way.

We will consider the case that $|\Theta| >> |\mathbb{T}|$, which implies that a topic word appears much more frequently in the sentences than a non-topic word.

**Attention Model** For this task, we consider a simple attention mechanism similar to the one proposed by Wang et al. (2016). Each word $w$ is associated with two parameters: an embedding $\nu_w \in \mathbb{R}^d$ and a key $\kappa_w \in \mathbb{R}^{d'}$. Based on a global query $q \in \mathbb{R}^{d'}$, the context vector of sentence $\chi$ is computed by $\bar{\nu}(\chi) = \sum_{w \in \chi} \nu_w \frac{\exp(q^T \kappa_w)}{Z(\chi)}$, where $Z(\chi) = \sum_{w' \in \chi} \exp(q^T \kappa_{w'})$. Then $\bar{\nu}(\chi)$ is fed into a classifier that predicts the sentence's topic in terms of a distribution over all topics.[1]

Denote the loss function by $l(\chi, y)$. Our upcoming analysis implies this attention model, although simple, may capture plenty of insight in understanding the training of more general attention models.

**Problem Statement** Our objective is to investigate the training dynamics, under gradient descent, of this attention model. In particular, we wish to understand if there is an intrinsic mechanism that allows the attention model to discover the topic word and accelerates training. Moreover, we wish to investigate, beyond this setup, how the model is optimized when there is no clear delimitation between topic and non-topic words, as in real-world data.

---

[1]The condition that the attention layer directly attends to the word embeddings merely serves to simplify the analysis in Section 4 but this condition is not required for most results presented in Sections 4 and 5. More discussions are given in Appendix A in this regard.

## 4 THEORETICAL ANALYSIS

It is common to fix some parameters when we train a model with limited resources. Also

**Lemma 1.** *Assume $q \neq 0$ when initialized. Fixing it does not affect the attention block's capacity.*

Thus, our upcoming discussion focuses on the case in which the query is fixed. Doing so also allows us to establish a closed-form expression connecting the word's embedding and the inner product of its key and the query. In Appendix B, extra discussions and experimental results reveal that the trainability of the query does not affect the fundamental relationship we are about to present.

For a topic word $t$, let $\Psi_t$ denote the training samples involving it. Then, by gradient descent,

$$\Delta \nu_t = \frac{\tau}{|\Psi|} \sum_{(\chi,y)\in\Psi_t} \nabla_{\bar{\nu}(\chi)} l(\chi,y) \ \frac{\exp(q^T \kappa_t)}{Z(\chi)} \tag{1}$$

$$\Delta \kappa_t = \frac{\tau}{|\Psi|} \sum_{(\chi,y)\in\Psi_t} q(\nu_t - \bar{\nu}(\chi))^T \ \nabla_{\bar{\nu}(\chi)} l(\chi,y) \ \frac{\exp(q^T \kappa_t)}{Z(\chi)}, \tag{2}$$

where $\tau$ denote the learning rate. As it will turn out, an important quantity in this setting is the inner product $q^T k_w$ of query $q$ and the key $k_w$, which we denote by $s_w$, and refer to it as the *score* of the word $w$.

Denoting $v_w = ||q||_2 \nu_w$, $\eta = \tau ||q||_2$, $\bar{v}(\chi) = \sum_{w\in\chi} \frac{\exp(s_w)}{Z} v_w$, and $h(\bar{v}(\chi); y) = \nabla_{\bar{\nu}(\chi)} l(\chi,y)$, for a topic word $t$, the dynamics simplifies to

$$\Delta v_t = \frac{\eta}{|\Psi|} \sum_{(\chi,y)\in\Psi_t} h(\bar{v}(\chi); y) \ \frac{\exp(s_t)}{Z(\chi)} \tag{3}$$

$$\Delta s_t = \frac{\eta}{|\Psi|} \sum_{(\chi,y)\in\Psi_t} (v_t - \bar{v}(\chi))^T \ h(\bar{v}(\chi); y) \ \frac{\exp(s_t)}{Z(\chi)}. \tag{4}$$

In the rest of the paper, whenever we refer to the embedding of word $t$, we actually mean $v_t$ not $\nu_t$.

Our analysis assumes the word embeddings are sampled i.i.d. from a distribution with mean zero and variance $\frac{\sigma^2}{d}$, where $\sigma^2$ is assumed close to zero. The word keys and the query are also sampled from zero mean distributions with a possibly different variance. We assume that this variance is so small that the initial word scores are approximately zero. This assumption of the initial configurations corresponds to the attention model starting as a word-averaging model, and allows us to investigate how the model deviates from this initial setting with training. We also assume the derivative $h(\bar{v}(\chi); y)$ of $\ell$ is Lipschitz continuous in $\bar{v}(\chi)$ throughout training. Further the assumption in Section 3 that the number of non-topic words $|\Theta|$ is much larger than the number of topic words $|\mathbb{T}|$ implies that with a sufficient number of training samples, the occurrence rate of a topic word is significantly higher than the non-topic ones. This then justifies the following assumption we will use throughout our analysis.

**Assumption 1.** *The scores and the embeddings of the non-topic words are nearly unchanged compared to their counterparts for the topic words.*

Hence, our upcoming analysis will treat the scores and embeddings of the non-topic words as constants. Assumption 1 will be validated by experimental results presented in Section 5.

By selecting a sufficiently small $\eta$, we can take the gradient-descent updates in Eq (3) and Eq (4) to its continuous-time limit and get[2]

$$\frac{\mathrm{d}v_t}{\mathrm{d}t} = \frac{\eta}{|\Psi|} \sum_{(\chi,y)\in\Psi_t} h(\bar{v}(\chi); y) \ \frac{\exp(s_t)}{Z(\chi)} \tag{5}$$

$$\frac{\mathrm{d}s_t}{\mathrm{d}t} = \frac{\eta}{|\Psi|} \sum_{(\chi,y)\in\Psi_t} (v_t - \bar{v}(\chi))^T \ h(\bar{v}(\chi); y) \ \frac{\exp(s_t)}{Z(\chi)}. \tag{6}$$

---

[2]Reversely, Eq (3) is a discretized approximation of Eq (5): $v_t(\mathrm{t}+1) - v_t(\mathrm{t}) = \int_t^{t+1} \frac{\mathrm{d}v_t(\mathrm{t}')}{\mathrm{d}t'} \mathrm{d}t' \approx 1 \cdot \frac{\mathrm{d}v_t(\mathrm{t})}{\mathrm{d}t} = \Delta v_t(\mathrm{t})$. The approximation becomes accurate if $v_t(\mathrm{t}+1)$ is close to $v_t(\mathrm{t})$, which can be achieved by choosing a sufficiently small $\eta$. Likewise, Eq (4) is a discretized approximation of Eq (6).

We can then characterize the update of the score and the embedding of a topic word as a continuous-time dynamical system stated in Lemma 2. The same technique has been used to analyze the training of neural networks in other contexts (Saxe et al., 2014; Greydanus et al., 2019).

**Lemma 2.** *For sufficiently small $\eta$ and $\sigma^2$, the score $s_t$ and embedding $v_t$ of topic word $t$ satisfy*

$$\frac{dv_t}{dt} = \frac{\eta|\Psi_t|}{|\Psi|} \left\langle h(\bar{v}(\chi); y) \frac{\exp(s_t)}{Z(\chi)} \right\rangle_{\Psi_t}, \tag{7}$$

$$\frac{ds_t}{dt} = \left[ (v_t - \langle \bar{v}(\chi \setminus t) \rangle_{\Psi_t})^T \frac{dv_t}{dt} \right] \left\langle \frac{\exp(s_t) + Z(\chi \setminus t)}{Z(\chi \setminus t)} \right\rangle_{\Psi_t}^{-1}, \tag{8}$$

*where $Z(\chi \setminus t) = \sum_{w \in \chi \setminus \{t\}} \exp(s_w)$, $\bar{v}(\chi \setminus t) = \sum_{w \in \chi \setminus \{t\}} v_w \frac{\exp(s_w)}{Z(\chi \setminus t)}$, and $\langle \cdot \rangle_{\Psi_t}$ denotes taking sample mean over the set $\Psi_t$.*

Eq (7) implies the speed of moving $v_t$ along the direction of $\langle h(\bar{v}(\chi); y) \rangle_{\Psi_t}$ is controlled by the attention weight $\frac{\exp(s_t)}{Z(\chi)}$. Eq (8) shows that $v_t$ increases if and only if $v_t$ has a greater projection on $\langle h(\bar{v}(\chi); y) \rangle_{\Psi_t}$ than the weighted average of the non-topic word counterparts.

Consider a simplified case where $\langle h(\bar{v}(\chi); y) \rangle_{\Psi_t}$ is fixed. Since the change of $v_t$ is much faster than the non-topic word counterparts, $v_t$ will have a larger projection on $\langle h(\bar{v}(\chi); y) \rangle_{\Psi_t}$ after a few epochs of training. Then $s_t$ increases as well as its attention weight, which in turn speeds up the extension of the embedding $v_t$. This observation reveals a mutual enhancement effect between the score increment and the embedding elongation. In fact such an effect exists in general, as stated in the theorem below, irrespective of whether $\langle h(\bar{v}(\chi); y) \rangle_{\Psi_t}$ is fixed.

**Theorem 1.** *In the setting of Lemma 2, from epoch $t_0$ to $t_1$, the topic word score $s_t$ and its embedding $v_t$ satisfy*

$$\left[ s_t(t) + \exp(s_t(t)) \left\langle \frac{1}{Z(\chi \setminus t)} \right\rangle_{\Psi_t} \right]_{t_0}^{t_1} = \left[ \frac{1}{2} ||v_t(t) - \langle \bar{v}(\chi \setminus t) \rangle_{\Psi_t} ||_2^2 \right]_{t_0}^{t_1}. \tag{9}$$

Following from Lemma 2, this theorem implies a positive relationship between the topic word score and the distance between $v_t$ and the non-topic word embedding average $\langle \bar{v}(\chi \setminus t) \rangle_{\Psi_t}$. Remarkably this result makes no reference to $\langle h(\bar{v}(\chi); y) \rangle_{\Psi_t}$, hence independent of it. This implies the identity in Eq (9) holds irrespective of the choice and setting of the classifier. Theorem 1 further implies a score and embedding norm ("SEN" in short) relationship for the topic words:

**Corollary 1.** *In the context of Theorem 1, by setting $t_0 = 0$ and $t_1 = t$, Eq (9) is reduced to*

$$||v_t(t)||_2 = \sqrt{2\left(s_t(t) + \frac{\exp s_t(t)}{m} - \frac{1}{m}\right)}, \tag{10}$$

The corollary indicates that $||v_t(t)||_2$ is monotonically increasing with $s_t(t)$. So, $s_t$ increases if and only if the point $v_t$ departs from its initial location. That is, if the norm of the topic word embedding increases, it will be attended to. This result is independent of the configuration of all other network layers. Thus, if $\langle h(\bar{v}(\chi); y) \rangle_{\Psi_t}$ has a gradient field that pushes $v_t$ away from its original location, the topic word is expected to be attended to. This statement can be made precise, as in Theorem 2, when the model uses a linear classifier.

**Theorem 2.** *Assume the model has a fixed classifier in the form $c(\bar{v}(\chi)) = softmax(U^T \bar{v}(\chi))$, where the columns of $U$ are linearly independent, and the model is trained using gradient descent with the cross-entropy loss. As training proceeds, the model will attend to the topic word in every input sentence and have its training loss approach zero.*

It is notable that the theorem holds broadly for any arbitrary fixed linear classifier (subjective to the mild linear independence constraint of its parameter $U$). Additionally, we anticipate that this result holds for a much wider family of classifiers including trainable and even nonlinear ones. But rigorous proof appears difficult to obtain in such settings, and we will corroborate this claim in an experimental study in Section 5.

To sum up, in this section, we have shown two main results: (a) there is a closed-form positive relationship, the SEN relationship, between the topic word score and its embedding norm, which is independent of the configuration of the classifier. (b) the model, equipped with a fixed linear classifier stated in Theorem 2, can be trained to have all topic words attended to.

# 5    EXPERIMENTAL STUDY

In this section, we first test our model on an artificial dataset generated through the procedure introduced in Section 3. The test corroborates our theoretical results and validates their assumptions. Our test results suggest that the attention mechanism introduces a synergy between the embedding and the score of topic words.

Another experiment is performed on the real datasets SST2 and SST5 (Socher et al., 2013). The experiment results suggest that the SEN relationship of topic words holds at least in initial training stages. As training proceeds, some words appear to deviate from the theoretical trajectories. Further analysis of this behaviour provides additional insights into the attention model's training dynamics on real-world datasets, often possessing a much more complex structure as well as rich noise. We performed all experiments using PyTorch (Paszke et al., 2017).

We performed our experiments on three models, Attn-FC, Attn-TC and Attn-TL, having the same attention block but different classifiers. The first two have the classifier in form $\mathbf{c}(\bar{v}(\chi)) = \text{softmax}(U^T \bar{v}(\chi))$ and the last in form $\mathbf{c}(\bar{v}(\chi)) = \text{softmax}(U_2^T \text{ReLu}(U_1^T \bar{v}(\chi) + b_1) + b_2)$. Except that the $U$ in Attn-FC is fixed, other parameters of the three models are trainable and optimized using the cross-entropy loss.

Since a real-world dataset does not have a topic word as the sentence topic indicator, we introduce a word "topic purity" measurement to facilitate our discussion on the experiments performed on SST2. Let $\delta_w^+$ and $\delta^-(w)$ respectively denote the portions of the positive and negative sentences among all training samples containing word $w$. Then the *topic purity* of $w$ is $\delta(w) = |\delta^+(w) - \delta^-(w)|$. If $\delta(w) = 1$, $w$ is either a pure positive or negative topic word. If $\delta(w) = 0$, $\delta^+(w) = \delta^-(w) = 0.5$, which implies $w$ has a completely random topic correspondence.

## 5.1    EXPERIMENTS ON SYNTHETIC DATASETS

The artificial dataset, consisting of 800 training and 200 test samples, is generated through the procedure introduced in Section 3. The dataset has four topics, and each contains two topic words. There are 20 non-topic words per sentence and the non-topic word dictionary size $M = 5,000$.

Our experiments use the same embedding dimension 15 for all three models. Regarding the classifiers, Attn-FC and Attn-TC adopt $U \in \mathbb{R}^{15 \times 4}$, while Attn-TL takes $U_1 \in \mathbb{R}^{15 \times 10}$, $b_1 \in \mathbb{R}^{10}$, $U_2 \in \mathbb{R}^{10 \times 4}$ and $b_2 \in \mathbb{R}^4$. For the validation of Theorem 2, the columns of $U$ in Attn-FC are set to be orthonormal and thus linear independent. Unless otherwise stated, the scores are set to zero and the embeddings are initialized by a normal distribution with mean zero and variance $\frac{\sigma^2}{d} = 10^{-6}$. We trained the models using gradient descent with learning rate $\eta = 0.1$ for 5K epochs before measuring their prediction accuracy on the test samples. When training is completed, all three models achieve the training loss close to zero and the $100.0\%$ test accuracy, which implies the trained models perfectly explain the training set's variations and have a good generalization on the test set.

**Verification of Assumption 1 and validation of Corollary 1 and Theorem 2.** We repeated the experiments for five runs and plotted the empirical score distributions of the non-topic and topic words of the three well-trained models with their $95\%$ confidence intervals in the first two graphs of Fig 1. Compared to the topic words, the scores of the non-topic words are nearly unchanged throughout the entire training process. Likewise, the next two plots show the embedding norms of the non-topic words are nearly constant, too. This implies Assumption 1 indeed holds. Fig 2 plots the empirical and the theoretical SEN curves of a randomly picked topic word for the three models, where the theoretical curve has the expression stated in Eq (10). The coincidence of the empirical and the theoretical curves in all three models validates the SEN relationship stated in Eq (10) and

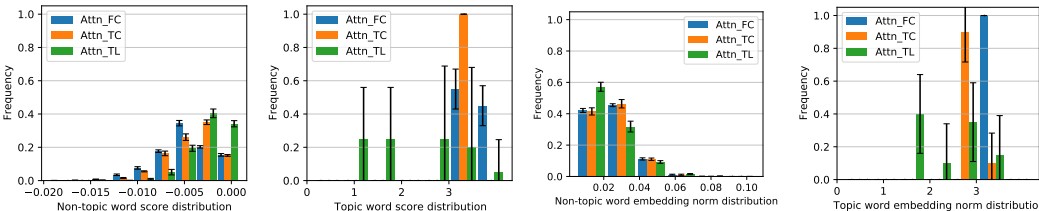

Figure 1: The first two histograms plot the distributions with the 95% confidence interval of the non-topic and topic word scores for the three well-trained models, Attn-FC, Attn-TC and Attn-TL by repeating the experiments five times. Likewise, the next two plot the ones of the embedding norms. The data of the models are overlaid in each histogram for easy comparisons.

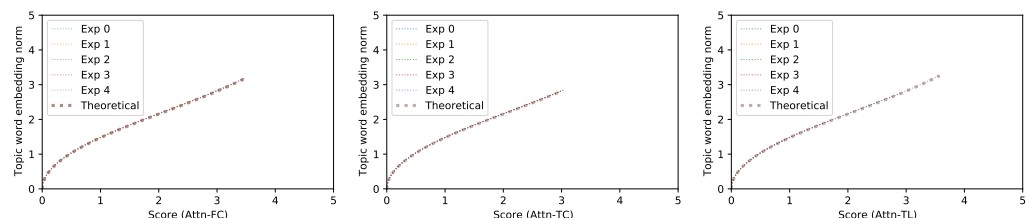

Figure 2: For Attn-FC, Attn-TC and Attn-TL, the graphs in order respectively plot the SEN relationship of a randomly picked topic word along with the theoretical predication derived in Corollary 1. For each model, we repeated the experiments five times and selected the same topic word.

its independence of the later layers.[3] Moreover, Fig 1 (left two) shows the scores of the topic words exceed the non-topic word counterparts by two orders of magnitude, which implies the topic words are attended to in a well-trained model. As we have reported earlier, Attn-FC has the training loss roughly zero when the training is completed, Theorem 2 is confirmed.

**Lower-capacity classifiers result in stronger attention effects.** The comparison, among the SEN distributions of the topic words in the three models, implies that a lower-capacity classifier leads to greater topic word SEN, which means a more drastic attention decision. This happens because the classifier of a larger capacity can explain more variations of the sample distribution and has more freedom to accommodate and absorb the correcting gradient signals. As a result, the attention layer receives a weaker gradient on average, which makes the embeddings of the topic word extend less from the original point. Thus, as implied by Eq (10), the magnitudes of the scores of the topic words are dampened, and therefore a weaker attention effect will be expected. This observation hints that if we know attending to the right words can explain most of the variations in sample distributions, we should consider a low capacity classifier (or later layers in general). Alternatively, we may also freeze the classifier in the initial stage of a training process, forcing the attention layer to explain more variations. Remarkably, all these modifications do not affect the relationship stated in Eq (10).

**Synergy between score growth and embedding elongation in topic words.** Eq (10) implies a positive SEN relationship of the topic word. That is, a larger topic word embedding norm results in a larger score (and thus a larger attention weight), which in turn makes the embedding extend faster. To corroborate this claim, we performed an ablation study by considering two variants of Attn-TC. The first has the scores frozen (referred as Attn-TC-KF) and the second has the embeddings fixed (referred as Attn-TC-EF). In this experiment, the embeddings of the models are initialized by a normal distribution of mean zero and variance $\frac{\sigma^2}{d} = 0.1$. We trained all three models by gradient descent for 60K epochs with learning rate $\eta = 0.1$, which are sufficient for all three models to fully converge. All three trained models reached 100% test accuracy.

---

[3]In Appendix B, the experiments with trainable queries are also implemented. The results indicate that the trainability of queries do not affect the positive SEN relationship. Besides, the query fixed model has very similar training dynamics to the one with a trainable query and a large initial norm.

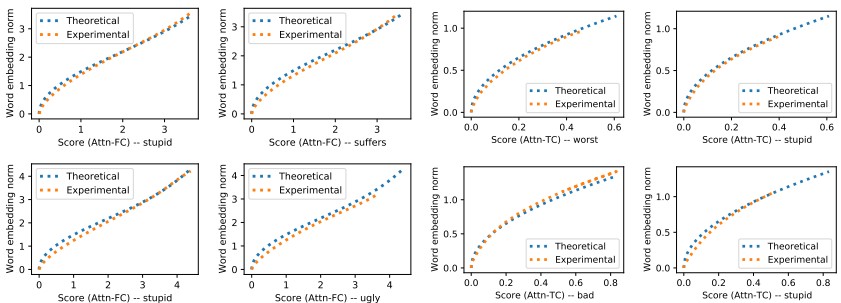

Figure 3: The changes of the SEN and the training loss for Attn-TC, Attn-TC-KF and Attn-TC-EF. The first two graphs demonstrate the evolution of a topic word SEN when the mutual enhancement happens. The third shows the change of the training loss. The last two show how the topic word SEN change in a separate training when the mutual diminution occurs.

Figure 4: The SEN relationship in the training process of Attn-FC (left two columns) and Attn-TC (right two columns) on SST2 (upper row) and SST5 (lower row). The words, picked among the ones having the largest five scores, have the SEN curves that agree with their theoretical counterparts.

The first three graphs of Fig 3 describe the evolution of the three models in the first 3K training epochs. For a randomly picked topic word, the first plot shows its score in Attn-TC grows faster than the one in Attn-TC-EF. Note that the score in Attn-TC-EF finally surpasses the one in Attn-TC because Attn-TC has converged at around 1K epochs. Likewise, the word embedding norm in Attn-TC increases more rapidly than the one in Attn-TC-KF before Attn-TC converges. The observations imply the attention introduces a mutual enhancement effect on training the topic word's score and its embedding, which makes Attn-TC enjoy the fastest training loss drop as shown in the third plot.

The mutual enhancement could become the mutual diminution in the early training stage if the initial embedding of the topic word has a negative projection on the direction that it will move along. This effect can be precisely characterized by Eq (8). Assume the embedding of a topic word is initialized to have a smaller projection on the gradient, passed from the classifier, than the average of the non-topic words. The reversed order of the projections makes the score of the topic word decrease as its embedding has a "negative effect" on the training loss compared to the average of the non-topic word embeddings. This will, in turn, impede the elongation of the embedding vector or even make it shrink (see the last two plots of Fig 3). The "negative effect" cannot last long because the topic word embedding moves along the gradient much faster than the non-topic words due to its high occurrence rate in the training samples. By Eq (8) again, $\frac{\mathrm{d}s_t}{\mathrm{d}t}$ will finally become positive. That is, the score of the topic word starts to increase and its attention weight will surpass the one of the word-averaging model (see the second last plot of Fig 3). Then, we start to observe the mutual enhancement effect, which is indicated in the last plot: the increase speed of the Attn-TC's embedding norm exceeds the Attn-TC-KF's since around the 370-th epoch.

## 5.2 EXPERIMENTS ON SST2 AND SST5

The second part of the experiment is performed on datasets SST2 and SST5, which contain movie comments and ratings (positive or negative in SST2 and one to five stars in SST5). For simplicity, we limit our discussion on Attn-FC and Attn-TC using the same configurations of our previous experiments except that the embedding dimension is set to 200. Remark that our goal is not to find a state-of-the-art algorithm but to verify our theoretical results and further investigate how an attention-based network works.

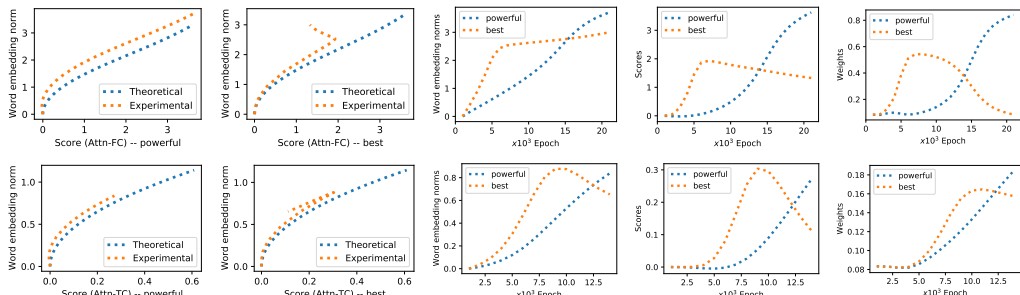

Figure 5: The juxtaposition of `powerful` and `best` during the training of Attn-FC (the first row) and Attn-TC (the second row). In each row, the first two graphs are the empirical and the theoretical SEN curves of words `powerful` and `best`. The next two plot their SEN as a function of the number of epochs. The last shows the changes of their weights in a sentence that they co-occur.

For both SST2 and SST5, we trained the two models by gradient descent with learning rate $\eta = 0.1$ combined with the early stopping technique (Prechelt, 2012) of patience 100. As PyTorch requires equal length sentences in a batch, we pad all the sentences to the same length and set the score of the padding symbol to the negative infinity. Under this configuration, the trained Attn-FC and Attn-TC reached $76.68\%$ and $79.59\%$ test accuracy on SST2 and $38.49\%$ and $40.53\%$ on SST5.

**Validation of Corollary 1.** As the true topic words are unknown in a real dataset, we checked the words of the largest fifty scores after the training is completed. We observed that most of the words have their SEN curves close to our theoretical prediction. We picked two words for each model-dataset combination and plotted the curves with their theoretical counterparts in Fig 4.

**The competition of two topic word candidates of various occurrence rates and topic purity.** To better understand how the attention block works, we investigated the case when the empirical and the theoretical SEN curves disagree. We limit our discussion on the model trained on SST2.

For both Attn-FC and Attn-TC, we noticed that there are mainly two types of deviations, as shown in the first two columns of Fig 5. In the first one, the theoretical curves overestimate the score growth of `powerful` in terms of the embedding norm, while `best` shown in the second column experiences a score drop combined with a failed theoretical prediction in the final training stage. These two types of the disagreement in fact occur in pairs, which is caused by a pair of words that one of them frequently appears in the training samples but has a low topic purity, while the other has the opposite.

Regarding the (`powerful`, `best`) pair, `best` appears in 128 training samples while 103 of them are positive. In comparison, `powerful` appears in the 36 training samples, and all of them are positive. The large difference in the number of samples makes the embedding norm of `best` extend much faster than `powerful` in the initial training process (Fig 5, third column). As the positive SEN relationship implies, a quicker embedding norm increase of `best` results in its faster score growth. Therefore, `best` will be more attended to ( Fig 5, last column), which thus further accelerates its SEN increase (this is the synergy relationship between the embedding and score that has been demonstrated in the 'ablation' study). This process does not stop until the gradient from the classifier diminishes because of its low topic purity (which is similar to the case when the label smoothing (Szegedy et al., 2016) is applied for alleviating the overfitting problem). In contrast, `powerful` is less trained initially due to its low occurrence rate. But its high topic purity makes the direction of the gradient stable, in which its embedding will steadily elongate. The elongation will, at last, let the embedding have a greater projection on the gradient vector than the average of the other words appearing in the same sentence. Thus, the score of `powerful` starts to increase as shown by Eq (8) and plotted in the second last column of Fig 5. In contrast, as the gradient magnitude drops, the embedding of `best` will extend in a decreasing speed; and its projection on the gradient, passed from the classifier, will finally be surpassed by the words co-occurring with it but having a higher topic purity (like `powerful`). Thus, its score starts to drop eventually.

**The dynamics of topic purity of attended words** The analysis of the inconsistency between the empirical and theoretical SEN curves hints that the disagreement are strongly related to the word's topic purity and its occurrence rate. To better characterize their dynamics, for a word $w$, let $\mathcal{A}_w(\mathtt{t})$

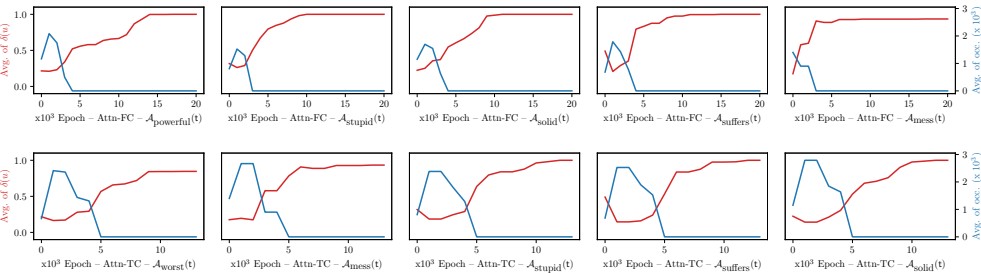

Figure 6: The average topic purity (red) and number of occurrence (blue) of the words attended to in the sentences containing $w$ as the training of Attn-FC (first row) and Attn-TC (second row) proceeds. Word $w$ is picked from the set of words having the largest five scores in the well-trained models.

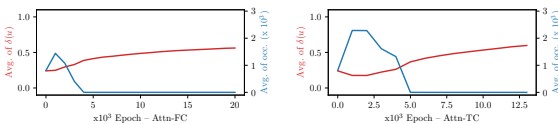

Figure 7: The average topic purity (red) and number of occurrence (blue) of the words attended to in all the training sentences as the training of Attn-FC (left) and Attn-TC (right) proceeds.

be the list, at epoch $t$, that records the attended word (or of the largest score) in the sentences containing $w$. If multiple words in a sentence have the same score, randomly pick one of them. Note that $\mathcal{A}_w(t)$ may contain repeating words as a word could be attended in multiple sentences. We selected $w$ to be the words having the largest five scores in the well-trained Attn-FC and Attn-TC, respectively. At various epoch $t$, Fig 6 plots how the average of topic purity $\delta(w')$ evolves for $w' \in \mathcal{A}_w(t)$ as well as the average number of occurrence in the training samples. For both models, they initially attend to the words that mostly have low topic purity with a high occurrence rate. As the training proceeds, the average topic purity of the attended words increases while the average occurrence rate drops. At the end of the training, almost all the attended words have a close-to-one topic purity. Fig 7 shows the evolution of the average topic purity and the average occurrence rate of the attended words over the entire training set. While a similar changing pattern can be observed, the average topic purity is lower than the one presented in Fig 6 when the training is completed. We argue that this happens because some sentences do not have any high topic purity words or their high topic purity words have a too low occurrence rate to be sufficiently trained.

## 6 CONCLUSION

This paper investigated the dynamic of training a series of attention-based bag-of-word classifiers on a simple artificial topic classification task. We have shown a persisting closed-form positive SEN relationship for the word to which the model should attend to. This result is independent of the configurations of the later layers. Through the result, we have proved that the model must converge in attending to the topic word with the training loss close to zero if the output of the attention layer is fed into a fixed linear classifier. A list of experiments has confirmed these results.

The experimental results indicate that the attention block intends to make a more drastic decision if its later layers have a lower capacity. This hints the classifier's limited capacity may help if "selecting" the right word explains most of the variations in the training samples. An ablation study shows a synergy between the topic word score's growth and its embedding elongation, leading to a faster training loss drop than the fixed score and fixed embedding variants. Besides, we have shown that this "mutual promotion" effect can also exhibit itself as "mutual suppression" in the initial training stage.

We investigated the competition of two topic word candidates with large differences in the topic purity and the occurrence rate in the training samples. The words of a higher occurrence rate but possibly low topic purity are more likely to be attended to initially. However, as the training proceeds, the attended words are gradually replaced by those of higher topic purity.

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

## A    THE RESULTS OF THE MODEL THAT THE ATTENTION BLOCK ATTENDS TO A CNN LAYER

The main text focuses on a model that has the attention block directly attends to word embeddings. Such design simplifies our analysis but should not be considered as a pre-condition to keep our results valid. In particular, the positive SEN relationship generally holds regardless of other components of the network, which can be justified as follows. There are two correlated sources of gradient signals, one back-propagates from the score to update key and query, the other back-propagates from the classifier loss to update word embedding. The correlation governs the positive relationship between score and embedding norm. Although the integration of other modules makes the analysis harder, the relationship should persist. Hence, all the results depending on this relationship keep valid.

We empirically verify our claims by implementing an experiment similar to the one discussed in the main text. We modified Attn-TC (named Attn-TC-CNN) by adding two parallel CNN layers to respectively process the word embeddings and the keys before feeding them into the attention block. Then, we construct an analogous data generation process introduced in Section 3. Finally, we empirically show that Assumption 1 and the positive SEN relationship still hold.

The Attn-TC-CNN has the same configurations as the Attn-TC (introduced in Section 5.1) except that we added two parallel CNN layers to preprocess the word embeddings and the keys. Regarding the CNN layer processing the word embeddings, it has the kernel of size $d \times 2$ and stride $1$. We used $d$ kernels to keep the word embedding dimension unchanged. So, for two consecutive words in a sentence, the CNN mixes their embeddings and produce a new one. Given that the sentence has $m + 1$ words, the input embedding matrix has shape $d \times (m + 1)$ and the output has shape $d \times m$. Likewise, regarding the keys, the CNN layer processing them has the kernel size $d' \times 2$ and stride $1$. And there are $d'$ kernels in total.

Consider a classification problem containing two topics $A$ and $B$. The sentences of the two topics are generated by two Markov chains, respectively, which are constructed as follows:

1. Let $L_i$ ($i = A, B$) be two mutually exclusive sets of ordered word pairs. The word pairs do not contain repeating words.
2. For $i = A, B$:

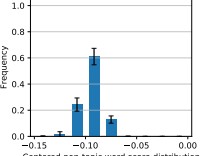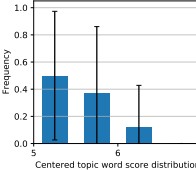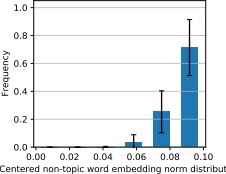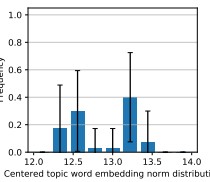

Figure 8: The first two histograms plot the distributions of the non-topic and topic word pair scores along with their 95% confidence intervals for the well-trained Attn-TC-CNN. Likewise, the next two plot the ones of the centered embedding norms.

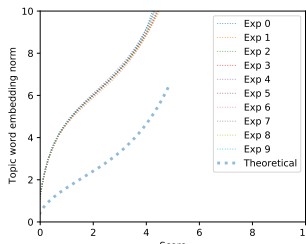

Figure 9: The empirical and theoretical SEN curves of a randomly picked topic word pair for model Attn-TC-CNN.

   (a) Initialize Markov Chain ($MC_i$) of words in the dictionary such that from a word, there is a uniform probability of moving to any words (including itself) in one step.

   (b) Group the word pairs in $L_i$ according to the leading words. For each group, let $s$ denote the shared leading word and $e_i$ ($i = 1, 2, \cdots, n$) the second. Set the probability of moving from $s$ to $e_i$ in one step be $\frac{1}{n}$ and those to the rest zero.

We call the word pairs in $L_i$ ($i = A, B$) the topic word pairs. For each pair of words, a new embedding and a new key are generated by feeding their original embeddings and keys into the CNNs. We refer to the new embedding as the topic word-pair embedding and the new key as the topic word-pair keys.

Likewise, for any other word pairs, the new generated embeddings and keys are referred to as the non-topic word-pair embeddings and keys.

Assume the training dataset contains sentences of length $m + 1$. We generate it by repeating the following procedure:

   1. uniformly sample a topic $i \in \{A, B\}$.

   2. uniformly sample a word pair from $L_i$ and let the first word be the starting word.

   3. sample another $m$ words by running the Markov process for $m$ times.

In our experiments, the word dictionary has 200 words, and so there are $40,000$ word pairs. The sentence length is set to 10. For $i = A, B$, each $L_i$ contains two randomly picked word pairs as the topic word pairs. Note that we have ensured that $L_A$ and $L_B$ are mutually exclusive. We used $4,000$ samples for training and $1,000$ for testing. The model was trained for $3,000$ epochs and achieved $99.89\%$ test accuracy.

We repeated the experiments for ten runs and plotted the distributions along with the 95% confidence interval of the scores and the embedding norms of the topic and the non-topic word pairs in Fig 8. Note that, the word pair embeddings and the keys are generated from the CNN layers. So the initial embeddings and the scores are not very closed to the origin. To facilitate the verification of Assumption 1, we centered them by subtracting their initial values before we plot their distributions. From Fig 8, we observe that the non-topic word pair scores and the embeddings are nearly unchanged in comparison to their counterparts of the topic ones. Therefore, we have shown that Assumption 1 is largely held even we have added CNN layers to process the word embeddings and the keys before feeding them into the attention block.

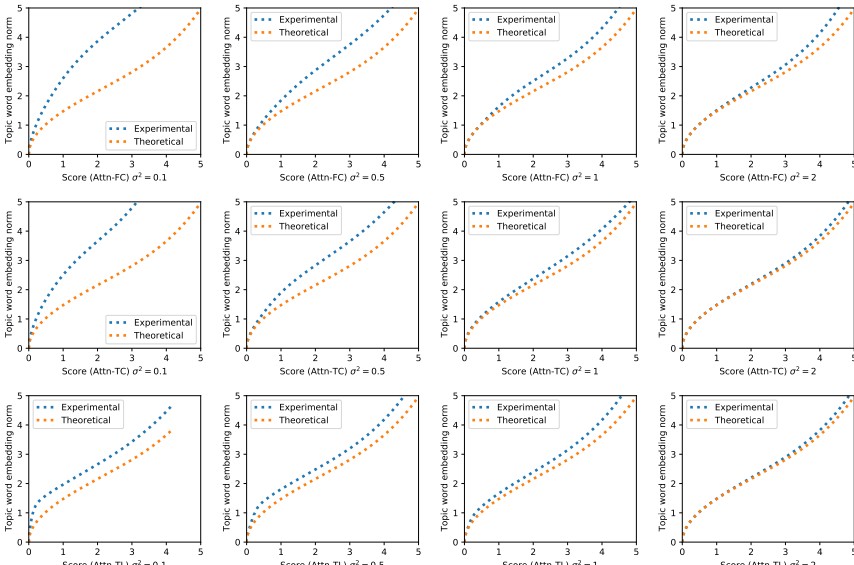

Figure 10: The empirical and the theoretical SEN curves of a randomly picked topic word for Attn-FC (first row), Attn-TC (second row) and Attn-TL (third row) with a trainable query. The query is initialized by a normal distribution $N(0, \sigma^2)$. From left to right, $\sigma^2 = 0.1, 0.5, 1$ and $2$, respectively.

Randomly picking a topic word pair, we plotted its empirical and theoretical SEN curves in ten runs in Fig 9. The figure shows that the positive SEN relationship holds even if the attention layer attends to other layers instead of the word embeddings directly. Therefore, all the results due to the relationship keep valid.

## B  DISCUSSION AND EXPERIMENTAL RESULTS OF THE MODELS WITH A TRAINABLE QUERY

In Section 4, we assumed a fixed and non-trainable query which allows the derivation of a clean closed-form "SEN" relation. But it is worth emphasizing that the positive relationship between the score and the embedding norm in fact exists regardless of the trainability of the query. As we have mentioned in Appendix A, there are two correlated sources of gradient signals, one back-propagates from the score to update key and query, the other back-propagates from the classifier loss to update word embedding. This correlation governs the positive relationship between score and embedding norm. Whether the query is trainable does not alter the existence of this correlation, although a trainable query makes the analysis more difficult. In particular, when the query norm is large, the update of query is relatively negligible; thence, the training behaviour is similar to having a fixed query.

To verify our claims, we reimplemented the experiments introduced in Section 5.1 with the same configurations except that the query is trainable. In Fig 10, we plot the empirical and the theoretical SEN curves of a randomly picked topic word by training Attn-FC, Attn-TC and Attn-FC with a trainable query. We initialize the entries of the query vector by a normal distribution $N(0, \sigma^2)$. From left to right, $\sigma^2$ increases as well as the initial query norm. We observe that regardless of how the query is initialized, the positive SEN relationship always preserves. Moreover, as the initial norm increases, the empirical curve approaches the theoretical curve having the expression in Eq (10). As we have discussed, this asymptotic approach happens since an increasing initial norm of the query makes its change negligible during the training process compared to its already big enough norm.

## C    PROOFS OF THE RESULTS IN SECTION 4

*Proof of Lemma 1.* Assume there are in total $N$ words in the dictionary (including both the topic and the non-topic words). Sample the keys of the $N$ words arbitrarily to generate $K \in \mathbb{R}^{d' \times N}$ with the key vectors as its columns. Randomly pick $q \in \mathbb{R}^{d'}$ as a query. To prove the lemma, it is sufficient to show for any non-zero $\tilde{q} \in \mathbb{R}^{d'}$, there exists $\tilde{K} \in \mathbb{R}^{d' \times N}$ such that $q^T K = \tilde{q}^T \tilde{K}$. Since $\tilde{q} \neq 0$, without loss of generality, assume its first entry $\tilde{q}_1$ is non-zero. Let $S = [s_1, s_2, \cdots s_N] = q^T K$. For $i = 1, 2, \cdots N$, let the $i$-th column of $\tilde{K}$ be $[\frac{s_i}{\tilde{q}_1}, 0, \cdots, 0]^T$. Then we can easily check that $q^T K = \tilde{q}^T \tilde{K}$. $\qquad \square$

*Proof of Lemma 2.* Picking a sufficiently small $\eta$, a continuous time limit can be taken to obtain the dynamics,

$$\frac{dv_t}{dt} = \frac{\eta}{|\Psi|} \sum_{(\chi,y) \in \Psi_t} h(\bar{v}(\chi); y) \frac{\exp(s_t)}{Z(\chi)}, \tag{11}$$

$$\frac{ds_t}{dt} = \frac{\eta}{|\Psi|} \sum_{(\chi,y) \in \Psi_t} (v_t - \bar{v}(\chi))^T \; h(\bar{v}(\chi); y) \frac{\exp(s_t)}{Z(\chi)}. \tag{12}$$

which are equivalent to

$$\frac{dv_t}{dt} = \frac{\eta|\Psi_t|}{|\Psi|} \left\langle h(\bar{v}(\chi); y) \frac{\exp(s_t)}{Z(\chi)} \right\rangle_{\Psi_t}, \tag{13}$$

$$\frac{ds_t}{dt} = \frac{\eta|\Psi_t|}{|\Psi|} \left\langle (v_t - \bar{v}(\chi))^T \; h(\bar{v}(\chi); y) \; \frac{\exp(s_t)}{Z(\chi)} \right\rangle_{\Psi_t}. \tag{14}$$

As $h(\bar{v}(\chi); y)$ is assumed Lipschitz continuous, we have[4]

$$\left| \langle v_t - \bar{v}(\chi) \rangle_{\Psi_t}^T \left\langle h(\bar{v}(\chi); y) \frac{\exp(v_t)}{Z(\chi)} \right\rangle_{\Psi_t} - \left\langle (v_t - \bar{v}(\chi))^T h(\bar{v}(\chi); y) \frac{\exp(v_t)}{Z(\chi)} \right\rangle_{\Psi_t} \right| < \frac{\mathcal{L}\sqrt{d}\sigma^2}{4}, \tag{15}$$

where $\mathcal{L}$ is the Lipschitz constant. Choosing a small enough $\sigma^2$ so that $\mathcal{L}\sqrt{d}\sigma^2$ is close to zero, we have

$$\langle v_t - \bar{v}(\chi) \rangle_{\Psi_t}^T \left\langle h(\bar{v}(\chi); y) \frac{\exp(v_t)}{Z(\chi)} \right\rangle_{\Psi_t} \approx \left\langle (v_t - \bar{v}(\chi))^T h(\bar{v}(\chi); y) \frac{\exp(v_t)}{Z(\chi)} \right\rangle_{\Psi_t}.$$

Then, combining it with Eq (14) yields

$$\frac{ds_t}{dt} = \frac{\eta|\Psi_t|}{|\Psi|} \left\langle (v_t - \bar{v}(\chi))^T \; h(\bar{v}(\chi); y) \; \frac{\exp(s_t)}{Z(\chi)} \right\rangle_{\Psi_t}$$

$$\approx \langle v_t - \bar{v}(\chi) \rangle_{\Psi_t}^T \frac{\eta|\Psi_t|}{|\Psi|} \left\langle h(\bar{v}(\chi); y) \frac{\exp(s_t)}{Z(\chi)} \right\rangle_{\Psi_t} = \langle v_t - \bar{v}(\chi) \rangle_{\Psi_t}^T \frac{dv_t}{dt}. \tag{16}$$

---

[4]The derivation is given in Appendix D

Remarkably, the relation stated in Eq (16) is independent of $h(\bar{v}(\chi); y)$. So the relationship does not depend on the architecture of the classifier (or later layers in general). Expand Eq (16),

$$\frac{\mathrm{d}s_t}{\mathrm{d}t} = \langle v_t - \bar{v}(\chi) \rangle_{\Psi_t}^T \frac{\mathrm{d}v_t}{\mathrm{d}t}$$

$$= \left\langle v_t - \left( \frac{\exp(s_t)}{Z(\chi)} v_t + \sum_{w \in \chi \setminus \{t\}} \frac{\exp(s_w)}{Z(\chi)} v_w \right) \right\rangle_{\Psi_t}^T \frac{\mathrm{d}v_t}{\mathrm{d}t}$$

$$= \left\langle \sum_{w \in \chi \setminus \{t\}} \frac{\exp(s_w)(v_t - v_w)}{Z(\chi)} \right\rangle_{\Psi_t}^T \frac{\mathrm{d}v_t}{\mathrm{d}t}$$

$$= \sum_{w \in \chi \setminus \{t\}} \left\langle \frac{\exp(s_w)}{Z(\chi)} \right\rangle_{\Psi_t} \langle v_t - v_w \rangle_{\Psi_t}^T \frac{\mathrm{d}v_t}{\mathrm{d}t}$$

$$\approx \sum_{w \in \chi \setminus \{t\}} \frac{\langle \exp(s_w)/Z(\chi \setminus t) \rangle_{\Psi_t}}{\langle Z(\chi)/Z(\chi \setminus t) \rangle_{\Psi_t}} \langle v_t - v_w \rangle_{\Psi_t}^T \frac{\mathrm{d}v_t}{\mathrm{d}t}.$$

The second last step is due to the independence between the score and embedding initialization, while the approximation in the last step is made as we assume all the scores of the non-topic words maintain the same during the entire training process. Rearranging the equation yields

$$\frac{\mathrm{d}s_t}{\mathrm{d}t} = \sum_{w \in \chi \setminus \{t\}} \left\langle \frac{\exp(s_w)}{Z(\chi \setminus t)} \right\rangle_{\Psi_t} \langle v_t - v_w \rangle_{\Psi_t}^T \frac{\mathrm{d}v_t}{\mathrm{d}t} \left\langle \frac{\exp(s_t) + Z(\chi \setminus t)}{Z(\chi \setminus t)} \right\rangle_{\Psi_t}^{-1}$$

$$= \left( v_t - \langle \bar{v}(\chi \setminus t) \rangle_{\Psi_t} \right)^T \frac{\mathrm{d}v_t}{\mathrm{d}t} \left\langle \frac{\exp(s_t) + Z(\chi \setminus t)}{Z(\chi \setminus t)} \right\rangle_{\Psi_t}^{-1}$$

where $\bar{v}(\chi \setminus t) = \sum_{w \in \chi \setminus \{t\}} v_w \frac{\exp(s_w)}{Z(\chi \setminus t)}$. □

*Proof of Theorem 1.* By Lemma 2, we have

$$\left\langle \frac{\exp(s_t) + Z(\chi \setminus t)}{Z(\chi \setminus t)} \right\rangle_{\Psi_t} \frac{\mathrm{d}s_t}{\mathrm{d}t} = \left( v_t - \langle \bar{v}(\chi \setminus t) \rangle_{\Psi_t} \right)^T \frac{\mathrm{d}v_t}{\mathrm{d}t}$$

which is

$$\left( 1 + \exp(s_t) \left\langle \frac{1}{Z(\chi \setminus t)} \right\rangle_{\Psi_t} \right) \frac{\mathrm{d}s_t}{\mathrm{d}t} = \left( v_t - \langle \bar{v}(\chi \setminus t) \rangle_{\Psi_t} \right)^T \frac{\mathrm{d}v_t}{\mathrm{d}t}. \tag{17}$$

By the fundamental theorem of calculus, integrating on both sides from $t = t_0$ to $t_1$ yields,

$$\left[ s_t + \exp(s_t) \left\langle \frac{1}{Z(\chi \setminus t)} \right\rangle_{\Psi_t} \right]_{t_0}^{t_1} = \left[ \frac{1}{2} || v_t - \langle \bar{v}(\chi \setminus t) \rangle_{\Psi_t} ||_2^2 \right]_{t_0}^{t_1}. \tag{18}$$

□

*Proof of Corollary 1.* Since the scores and the embeddings of the non-topic words are considered constant, we have $\left\langle \frac{1}{Z(\chi \setminus t)} \right\rangle_{\Psi_t} = \frac{1}{m}$ and $\langle \bar{v}(\chi \setminus t) \rangle_{\Psi_t} = 0$. As $v_t$ is initialized with mean zero and a very small variance, $||v_t(0)||_2^2 \approx 0$. Then, Eq (9) can be written as

$$||v_t(t)||_2 = \sqrt{2 \left( s_t(t) + \frac{\exp s_t(t)}{m} - \frac{1}{m} \right)}.$$

□

*Proof of Theorem 2 (sketch).* Without loss of generality, pick topic word $t$ and assume it corresponds to the $\varphi$-th topic. We prove the theorem by showing that as the number of epochs increases, for any sentence $\chi$ in $\Psi_t$, $s_t \to \infty$ and $\text{softmax}(U^T \bar{v}(\chi)) \to e_\varphi$, where $e_\varphi$ is the one-hot vector that the $\varphi$-th entry equals one and the rest are zeros. Let $x = U^T \bar{v}(\chi)$. Notice that the loss function $\langle -\log(\text{softmax}_\varphi(x)) \rangle_{\Psi_t}$ is convex in terms of $x$. As $U$ is fixed, the loss function is also convex in terms of $\bar{v}(\chi)$. This implies, if the model is optimized by gradient descent, the gradient will lead $\bar{v}(\chi)$ to its optimal solution $\bar{v}^*(\chi)$. In our case, as the columns of $U$ are linearly independent, there exists a vector $\mathbf{n}$ that are orthogonal to all the columns of $U$ except $U_\varphi$. Without loss of generality, assume $U_\varphi \cdot \mathbf{n} > 0$ (otherwise, choose its inverse). Then a potential optimal solution is $\bar{v}^*(\chi) = \lambda \mathbf{n}$ for $\lambda$ goes to the infinity as $\bar{v}^*(\chi) \cdot U_i = 0$ for $i \neq \varphi$ and $\bar{v}^*(\chi) \cdot U_\varphi \to \infty$, which implies $\text{softmax}(U^T \bar{v}^*(\chi)) = e_\varphi$. Combined with the fact that there cannot be an optimal solution $v^{**}(\chi)$ of a finite norm such that $\text{softmax}(U^T \bar{v}^{**}(\chi)) = e_\varphi$, the gradient must lead $\bar{v}(\chi)$ to an optimal solution that is arbitrarily far away from the origin, which also applies to $v_t$ as it receives the same gradient up to a multiple according to Eq (7). As $||v_t||_2^2$ increases unboundedly, by Theorem 1, the score $s_t \to \infty$ as well. So we have $\text{softmax}(U^T \bar{v}(\chi)) \to \text{softmax}(U^T v_t) \to e_\varphi$ and thus the cross-entropy loss drops to zero. $\qquad \square$

## D  THE DERIVATION OF EQ (15)

For a vector $u$, let $u^{(\varphi)}$ denote its $\varphi$-th entry. As we ignore the changes of the scores and the embeddings of non-topic words, their distributions maintain the same as the initial ones. In particular, the scores of the non-topic words are always zero. So, for $\varphi = 1, 2, \cdots d$,

$$\text{var}\left(\bar{v}^{(\varphi)}(\chi)\right) = \text{var}\left(\frac{\exp(s_t)}{Z(\chi)}v_t^{(\varphi)} + \sum_{w \in \chi \setminus \{t\}} \frac{\exp(s_w)}{Z(\chi)}v_w^{(\varphi)}\right)$$

$$= \sum_{w \in \chi \setminus \{t\}} \left(\frac{\exp(s_w)}{Z(\chi)}\right)^2 \frac{\sigma^2}{d} = \frac{m\sigma^2}{(Z(\chi))^2 d}.$$

Since $h(\bar{v}(\chi); y)$ is assumed Lipschitz continuous in $\bar{v}(\chi)$, there exists $\mathcal{L} \in \mathbb{R}$ such that for $\varphi = 1, 2, \cdots d$,

$$|h(\bar{v}(\chi_1); y)^{(\varphi)} - h(\bar{v}(\chi_2); y)^{(\varphi)}| \leq \mathcal{L}||\bar{v}(\chi_1) - \bar{v}(\chi_2)||_1,$$

where $\bar{v}(\chi_1), \bar{v}(\chi_2) \in \mathbb{R}^d$ and $||\_||_1$ denote the $l^1$-distance by taking the sum of the absolute values of the entry differences on each dimension. So we also have

$$\left|\frac{1}{\mathcal{L}}h(\bar{v}(\chi_1); y)^{(\varphi)} - \frac{1}{\mathcal{L}}h(\bar{v}(\chi_2); y)^{(\varphi)}\right| \leq ||\bar{v}(\chi_1) - \bar{v}(\chi_2)||_1.$$

According to the work of Bobkov & Houdrè (1996), for $\varphi = 1, 2, \cdots, d$,

$$\text{var}(\mathcal{L}^{-1}h(\bar{v}(\chi); y)^{(\varphi)}) \leq \sum_{\varphi=1}^d \text{var}\left(\bar{v}^{(\varphi)}(\chi)\right) = \frac{m\sigma^2}{(Z(\chi))^2},$$

which is

$$\text{var}(h(\bar{v}(\chi); y)^{(\varphi)}) \leq \frac{m\sigma^2 \mathcal{L}^2}{(Z(\chi))^2}.$$

Then, the Cauchy-Schwarz inequality implies, for $\varphi = 1, 2, \cdots, d$,

$$\left|\text{cov}\left(h(\bar{v}(\chi); y)^{(\varphi)}\frac{\exp(s_t)}{Z(\chi)}, v_t^{(\varphi)} - \bar{v}^{(\varphi)}(\chi)\right)\right| = \left|\text{cov}\left(h(\bar{v}(\chi); y)^{(\varphi)}\frac{\exp(s_t)}{Z(\chi)}, \bar{v}^{(\varphi)}(\chi)\right)\right|$$

$$\leq \left[\text{var}\left(h(\bar{v}(\chi); y)^{(\varphi)}\frac{\exp(s_t)}{Z(\chi)}\right)\right]^{1/2} \left[\text{var}\left(\bar{v}^{(\varphi)}(\chi)\right)\right]^{1/2} < \frac{m\sigma^2 \mathcal{L}\exp(s_t)}{(Z(\chi))^3 \sqrt{d}} \qquad (19)$$

By the triangle inequality,

$$
\left| \langle v_t - \bar{v}(\chi) \rangle_{\Psi_t}^T \left\langle h(\bar{v}(\chi); y) \frac{\exp(s_t)}{Z(\chi)} \right\rangle_{\Psi_t} - \left\langle (v_t - \bar{v}(\chi))^T h(\bar{v}(\chi); y) \frac{\exp(s_t)}{Z(\chi)} \right\rangle_{\Psi_t} \right|
$$

$$
= \left| \sum_{\varphi=1}^{d} \left[ \left\langle v_t^{(\varphi)} - \bar{v}^{(\varphi)}(\chi) \right\rangle_{\Psi_t} \left\langle h(\bar{v}(\chi); y)^{(\varphi)} \frac{\exp(s_t)}{Z(\chi)} \right\rangle_{\Psi_t} - \left\langle (v_t^{(\varphi)} - \bar{v}^{(\varphi)}(\chi)) h(\bar{v}(\chi); y)^{(\varphi)} \frac{\exp(s_t)}{Z(\chi)} \right\rangle_{\Psi_t} \right] \right|
$$

$$
\leq \sum_{\varphi=1}^{d} \left| \left\langle v_t^{(\varphi)} - \bar{v}^{(\varphi)}(\chi) \right\rangle_{\Psi_t} \left\langle h(\bar{v}(\chi); y)^{(\varphi)} \frac{\exp(s_t)}{Z(\chi)} \right\rangle_{\Psi_t} - \left\langle (v_t^{(\varphi)} - \bar{v}^{(\varphi)}(\chi)) h(\bar{v}(\chi); y)^{(\varphi)} \frac{\exp(s_t)}{Z(\chi)} \right\rangle_{\Psi_t} \right|
$$

$$
= \sum_{\varphi=1}^{d} \left| \mathrm{cov} \left( h(\bar{v}(\chi); y)^{(\varphi)} \frac{\exp(s_t)}{Z(\chi)}, v_t^{(\varphi)} - \bar{v}^{(\varphi)}(\chi) \right) \right|
$$

$$
< \frac{m \sigma^2 \mathcal{L} \exp(s_t) \sqrt{d}}{(Z(\chi))^3} \qquad \text{by Eq (19)}
$$

$$
= \frac{m}{Z(\chi)} \cdot \frac{\exp(s_t)}{Z(\chi)} \cdot \frac{\mathcal{L}\sqrt{d}\sigma^2}{Z(\chi)}
$$

$$
= \left( 1 - \frac{\exp(s_t)}{Z(\chi)} \right) \cdot \frac{\exp(s_t)}{Z(\chi)} \cdot \frac{\mathcal{L}\sqrt{d}\sigma^2}{Z(\chi)}
$$

$$
\leq \frac{\mathcal{L}\sqrt{d}\sigma^2}{4 Z(\chi)}
$$

$$
\leq \frac{\mathcal{L}\sqrt{d}\sigma^2}{4}.
$$

The last line is due to $Z(\chi) > \sum_{w \in \chi \setminus \{t\}} \exp(s_w) = m \geq 1$.

