# OpenReview forum: "On the Dynamics of Training Attention Models"
_ICLR.cc/2021/Conference — ICLR 2021 Poster_

### Official Review · AnonReviewer4 · 2020-10-15
**A potentially valuable theoretical starting point, requiring significant assumptions.**

**Rating:** 8
**Confidence:** 3

**Review:**

Summary:

This paper aims to prove and illustrate that attention components are defined during training by gradients that mutually amplify the embedding and score associated with crucial features. In particular, a word embedding with a high magnitude increases the gradient following the attention score for the same word, while a high attention score increases the gradient directed at the word's embedding. In addition to a proof that treats behavior during training as a dynamical system under a large suite of assumptions, they test the analytic predictions on a synthetic dataset following the same suite of assumptions. They then test on a natural language data set and discuss where it diverges from the analytic and synthetic findings,  concluding that the difference is a result of competition between different words associated with a label.

Pros:
1. We currently lack any substantial theory about attention modules and why they work. Although their model is simplistic, it could provide essential groundwork for analytic understanding of these popular systems. I would even consider it fairly realistic relative to a lot of the assumptions required for theoretical results in training dynamics research. Currently theory of attention is grounded in infinite-width networks, an assumption this paper does not make.
2. The synthetic results appear to substantiate this theoretical result.
3. They find an interesting result that, in more realistic settings, the learning dynamics follow particular patterns on the words that are paired together with more versus less predictive words. The framing of these effects in terms of competition between possible topic words is clearly inspired by considering which assumptions behind their proof have failed, which is evidence that the thinking behind their proof is potentially valuable.

Cons:
1. It's not clear how dynamics like these would generalize to multilayer attention networks like BERT.
2. The assumptions behind the theoretical and synthetic empirical results are simplistic:  The existence of a large vocabulary of "non-topic" words required to keep the variance of embedding negligible in out of focus words, the presence of only one topic word. There is also the very common assumption of Lipschltz continuity.
3. The natural language experiments make a specific claim about the different dynamics for competing words of different topic purity, but only presents an example of two words as evidence. I want to see quantitative evidence of the pattern.
4. The synthetic results would be strengthened by including multiple runs with different initializations so they can include confidence intervals.

 Questions:
1. Does this mutual amplification effect have any ramifications for the debate over whether attention weights can be used as a proxy for saliency?
2. In Lemma 1, there is a reference to the attention block's capacity which is difficult to decipher. What do you mean here by capacity?
3. The assumption that word embeddings are sampled from a distribution with small variance seems likely to apply early in training, but not later. Have you checked the actual variance that would be associated with word embeddings late in training?
4. What is actually meant by a word being "paired" with another word in the natural language experiments?
5. Did I misunderstand something in interpreting gradients amplifying the embedding and score as directed towards v and k respectively in this simplified model?

 Minor:
1. Notation is difficult to follow at times because several unrelated concepts use almost the same symbols: $s_i$ indicates score, but $S_i$ indicates a sentence; $\tau$ indicates learning rate, but $\mathcal{T}$ (which looks identical as a subscript) indicates a set of sentences.
2. In discussing early alignment of attention to syntax, Clark et al. 2019 was concurrent with https://www.aclweb.org/anthology/P19-1580/

---

> ### Author Response · Authors · 2020-11-13
> **Clarification**
>
>
> We thank you for your time and constructive suggestions. We are particularly pleased that you appreciate research of such a style. We have carefully gone through all your comments and suggestions and have the following responses, explanations and modifications.
>
>
> - *It's not clear how dynamics like these would generalize to multilayer attention networks like BERT.*
>
> 	**Response**: Our results are unlikely to generalize to BERT directly.  The transformer model in BERT not only has multiple attention layers, but also having multiple queries in each layer. We are still in the process of understanding such single-layer multi-query self attention models. In fact, we have noticed some interesting behaviour of the model, which appears to depend on how word embedding dimension relates to the number of labels in the classification tasks. We are in the process of formulating this behaviour, in a separate work.
>
>
> - *The assumptions behind the theoretical and synthetic empirical results are simplistic:  The existence of a large vocabulary of "non-topic" words required to keep the variance of embedding negligible in out of focus words, the presence of only one topic word. There is also the very common assumption of Lipschltz continuity.*
>
> 	**Response**: We agree that our setting is simple. Nonetheless, we consider the insights obtained in this work valuable.
>
> - *The natural language experiments make a specific claim about the different dynamics for competing words of different topic purity, but only presents an example of two words as evidence. I want to see quantitative evidence of the pattern.*
>
> 	**Response**: In fact, our paper has also provided the quantitative evidence in a macro-scale. Fig 6 demonstrates the changes in the topic purity and the occurrence rate of the most attended words. In this way, we see how the most attended words interact with the other less attended ones in a sentence and how the attended words are gradually replaced by those with higher topic purity but less occurrence rate. Considering that Fig 6 focuses on the sentence set that contains a specific word, we also provide extra results in Section C.3, focusing on all the training sentences. We hope this addresses your concerns, and we are glad to offer extra results if needed.
>
>
> - *The synthetic results would be strengthened by including multiple runs with different initializations so they can include confidence intervals.*
>
> 	**Response**: We agree with your comments and have added a modified Fig 1 in Section C.4. We decided not to add confidence intervals in Fig 2 because, in different experiments, neither the topic word embedding norms nor the scores have the same values, which makes the confidence interval not applicable. Instead, for each model, we repeated the experiments five times and plotted the experimental SEN curves of the same topic word (see Fig 12 in Section C.4.). We will use the modified figures to replace Fig 1 and Fig 2 in the revised paper.
>
>
> - *Does this mutual amplification effect have any ramifications for the debate over whether attention weights can be used as a proxy for saliency?*
>
> 	**Response**: Yes. The mutual amplification effect implies that the word with a faster embedding elongation will be assigned with a higher weight. Also, a fast word embedding extension hints the training of the word accounts for most of the training loss drop in comparison to the others. Thus, if we think a word saliency is related to its capability of causing a faster training loss drop, then the attention assigns the weights based on the word's importance and can be used as a proxy for saliency.
>
>
>
> - *In Lemma 1, there is a reference to the attention block's capacity which is difficult to decipher. What do you mean here by capacity?*
>
> 	**Response**: When two models contain precisely the same family of hypotheses, we say they have the same capacity.
>
> - *The assumption that word embeddings are sampled from a distribution with small variance seems likely to apply early in training, but not later. Have you checked the actual variance that would be associated with word embeddings late in training?*
>
> 	**Response**: Our analysis only requires that the non-topic word maintains the word embeddings of a small variance. In the experiment on the artificial data introduced in Section 5.1, if the embedding is initialized with variance 0.01, then the sample variances of all random word embeddings on a randomly selected dimension are: 0.01005027, 0.01005502, 0.01007887, 0.01008423, 0.01008448 at the 1K-th, 2K-th, ..., 5K-th epochs. We can observe that the variance increases at a very low and decreasing speed. So, the assumption is well held.

---

> > ### Author Response · Authors · 2020-11-13
> > **Clarification (cont.)**
> >
> > - *What is actually meant by a word being "paired" with another word in the natural language experiments?*
> >
> > 	**Response**: In that context, we merely emphasize that there are two types of inconsistency between the empirical and the theoretical curves, as shown in Fig 5. That is, if we observe one word A with one type of the SEN curve inconsistency, there may exist another word B frequently co-occurring in some sentences with A that has the other type of inconsistency.
> >
> > - *Did I misunderstand something in interpreting gradients amplifying the embedding and score as directed towards v and k respectively in this simplified model?*
> >
> > 	**Response**: No. Your understanding is precise.
> >
> >
> > - *Notation is difficult to follow at times because several unrelated concepts use almost the same symbols*
> >
> > 	**Response**: We will make the corresponding changes in the revised version.
> >
> > - *In discussing early alignment of attention to syntax, Clark et al. 2019 was concurrent with https://www.aclweb.org/anthology/P19-1580/*
> >
> > 	**Response**: Thank you for pointing to this reference. It is certainly related to the scope of this work. We will include and discuss it in the related work section in the revised version.

---

> > ### Comment · AnonReviewer4 · 2020-11-19
> > **attention weights as a proxy for saliency**
> >
> > The connection to this debate is interesting enough to lead to further work, building on the paper's theory in more complex settings and modifying it for fewer assumptions. (Again, existing published theory on attention requires extremely unrealistic assumptions already in different directions.) This introduces possible theoretical explanations for how they succeed and fail compared to more general interpretability methods, which has potential impact for a debate that is currently based exclusively on empirical results.

---

> > > ### Author Response · Authors · 2020-11-20
> > > **Thank you very much**
> > >
> > > We very much appreciate this insightful comment!

---

### Official Review · AnonReviewer1 · 2020-10-28

**Rating:** 6
**Confidence:** 3

**Review:**

This paper provides theoretical insight into the mechanisms by which a simplified attention model trained with gradient descent learns to allocate more mass to relevant words in the input.

In a limited toy setting, the authors derive a closed form relationship between word-score and word embedding norm in a simplified, one layer attention model. The theoretical findings are verified empirically both on the toy task and on a more realistic sentiment classification benchmark. Due to the extreme simplicity of the setting considered, as well as the number of assumptions made, it is unclear to me what to make of these results. In particular, it seems that the setting considered (fixed query attention over bag of word embeddings) is very different from real use cases of attention.

**Pros**
- The closed-form relationship between attention score and embedding norm during SGD training is novel as far as I know
- The theoretical results are well justified in experiments: in particular the predicted "SEN" relationship seems to match the prediction.


**Cons**
- Large number of assumptions, the validity of which is unclear in practice: in particular
   1. The assumption that the query vector is (1) a parameter and not a function of the inputs (as in self attention or cross-sentence attention) and (2) is fixed. I don't know of many "real world" attention networks that work this way, after all one of the main appeals of attention is its "content-based" nature
   2. Assumption 1 that the score and embeddings of non-topic words don't change during training. First, this seems like something that could be proven from the earlier assumption that the topic words are updated more frequently. And second it is unclear if it holds for a real task (and a different model where eg. the attention layer attends to higher layers rather than just the embeddings)
- Confusing notation makes the paper hard to follow (see remarks for examples)
- Unclear takeaway: what does this paper tell us about attention as it is used in practice?


**Remarks**
-  5.1: "The “negative effect” cannot last long because the topic word embedding moves along the gradient much faster than the non-topic words due to its high occurrence rate": This is true in the toy example in the paper, but is this the case in practice? For instance in sentiment classifications there are many words to describe sentiment that are infrequent (cue Zipf's law). Moreover, in realistic settings there will be non-topic words which appear very frequently (stop words such as "the", "a" in English).
- Lemma 1: while it is true that fixing q doesn't change the capacity of the model, it will definitely change its training dynamics (which is very much the theme of the paper as per the title). How important is it to fix q from this perspective?
- A lot of the math would be easier to read if the dependence of some variables (\hat v, Z,...) on a specific sentence was marked explicitly (eg. Z_S instead of Z)
- The notation in Lemma 2 was extremely confusing to me, due to the sudden introduction of the bracket notation and the awkward spacing with both equations on the same line. I would recommend at least putting both on separate line, and also reorganizing so that the LHS of the second equation is only ds_i/dt (move the mean to the other side)
- In 3. I think using "\mathbf R" for the dictionary is unfortunate (too similar to \mathbb R). Overall I found the separation between topic and non-topic words dictionaries confusing. Why not have a global Vocabulary V, a set of topic words T and refer to the remaining words as V\T?
- In 2. "[Hahn and Brunner] show theoretically that the self-attention blocks are severely limited in their computational capabilities when the input sequence is long, which implies that self-attention may not provide the interpretability that one expects.": can you clarify this sentence? Limitation in computational capabilities does not seem to entail limited interpretability in general (see linear models for instance).
- Typo in citations in 2.: "Hahn (Hahn, 2020) and Brunner (Brunner et al., 2019)" -> "Hahn (2020) and Brunner et al. (2019)"
- Typo in 3. "The training set [...] are" -> "The training set [...] is"

---

**Post Rebuttal**

In my review, the main concerns were (1) validity of assumptions, (2) confusing writing/notation and (3) unclear takeaway. The rebuttal appropriately addressed (1), although I am looking forward to the revision to see how this is discussed in the paper itself. I cannot really say anything about any improvements on the writing (2) without seeing the revision, but I am confident that the authors can address most of the issues pointed out by myself and other reviewers. Regarding (3), unclear takeaway, after reading the authors' response as well as the other reviews, my concerns are somewhat assuaged (partly because the assumptions were addressed better), although I am still unsure how or if the results in this paper could be expanded to realistic attention models.

There are additional issues I raised during the discussion (general lack of citations in particular), however this can be fixed fairly easily for the camera ready so I am willing to give the benefit of doubt and raise my score to 6 (borderline accept)

---

> ### Author Response · Authors · 2020-11-13
> **Clarification**
>
> Thank you for your reviews! We have carefully gone through them, and here are our responses, explanations and additional results.
>
>
> - *Large number of assumptions, the validity of which is unclear in practice: in particular*
>
> 1. *The assumption that the query vector*
>
> 	(1) *is a parameter and not a function of the inputs (as in self attention or cross-sentence attention) and (2) is fixed.*
>
> 	**Response**:
> Regarding your complaint that the query in our setting is not a function of the input, we agree that in many modern attention-based models (e.g, transformers), this is not the case. But in the simple setting of our studied task, there is no benefit to consider the query dependent of the input sentence. This is because this task boils down extracting topic words from each sentence and such extraction requires no context information due to the simplicity of the problem set up. We agree that the next step of this research should scale up the task complexity so that the tasks are closer to real-world applications.
>
> Having said this, in earlier attention-based models, indeed there exist abundant models (mostly for simpler tasks like text classification) in which query is chosen as a parameter independent of the input sentence, see, e.g., Wang et al, “Attention-based LSTM for Aspect-level Sentiment Classification”, ACL 2016. (Google Scholar Citations 830).
>
> Regarding your complaint that the query is made fixed and non-trainable, we first note that this choice allows the derivation of a clean closed-form "SEN" relation. But it is worth emphasizing that the positive relationship between the score and the embedding norm in fact exists regardless of the trainability of the query. This can be seen as follows. There are two correlated sources of gradient signals, one back-propagates from the score to update key and query, the other backpropagates from the classifier loss to update word embedding. This correlation governs the positive relationship between score and embedding norm. Whether the query is trainable does not alter the existence of this correlation, although trainable query makes the analysis more difficult. In particular, when the query norm is large, the update of query is relatively negligible; as a consequence, the training behavior is similar to having a fixed query.
> To support our claims, we have provided extra experimental results in Section C.1.
>
> In summary, the conclusion of this paper is valid even for the case when the query is trainable.
>
> 2. *Assumption 1 that the score and embeddings of non-topic words don't change during training.*
> 	- *First, this seems like something that could be proven from the earlier assumption that the topic words are updated more frequently.*
> 	**Response**: Yes. It is possible to show analytically that the score and embedding for topic words are NEARLY unchanged during training. This justifies Assumption 1. But in Assumption 1, the word “nearly” is removed, hence we call it an assumption.
>
> 	- And second it is unclear if it holds for a real task (and a different model where eg. the attention layer attends to higher layers rather than just the embeddings)*
> 	**Response**: There should be a much larger family of models and tasks in which the assumption holds. To support this claim, we have provided more experimental results in Section C.2. In particular, we have discussed a variant of the Attn-TC model (named Attn-TC-CNN) that contains a CNN layer to preprocess the word embeddings and the keys before feeding them into the attention block. The empirical results show that Assumption 1 still largely holds in this case.
> 	Regarding the assumption's validity in a real task, it is very hard to check it directly as the true topic words are unknown and the tasks may possess a more complex structure. Nonetheless, as we have presented in Section 5.2, the theoretical SEN curves on real datasets largely coincide with the empirical counterparts. This to an extent supports the assumption,  which we used to derive the expression of the theoretical SEN curve.

---

> > ### Author Response · Authors · 2020-11-13
> > **Clarification (cont.)**
> >
> >
> >
> > - *Unclear takeaway: what does this paper tell us about attention as it is used in practice?*
> >
> > 	 **Response**: This makes advances our understanding of attention mechanisms. In particular, we show a positive correlation between score and word embedding norm, which in turn justifies the effectiveness of attention in accelerating training. These results have provided some explanation of the working of attention models, despite the simple setting of this paper. The solid progress we made in understanding attention may serve as a ground for further study of attention in more complicated network architectures and for more complex tasks.
> >
> > Our experiments also provide valuable insights. For example,  a stronger attention effect can be gained by using a classifier of smaller capacity; the competition effects are likely to play important roles in the training dynamics of attention models in practical settings. Further study of this latter aspect may produce interesting discoveries.
> >
> >
> > - *"The "negative effect" cannot last long because the topic word embedding moves along the gradient much faster than the non-topic words due to its high occurrence rate": This is true in the toy example in the paper, but is this the case in practice? For instance in sentiment classifications there are many words to describe sentiment that are infrequent (cue Zipf's law). Moreover, in realistic settings there will be non-topic words which appear very frequently (stop words such as "the", "a" in English).*
> >
> > 	**Response**:  Practical tasks diverge from the studied setting in many ways. If we restrict to the studied setting allowing each non-topic word to have different occurrence frequencies so that some non-topic words occur with high frequency, the gradient signals these words receive in positive and negative examples cancel each other. This would make the training dynamics similar to the studied setting. If some topic words appear with low frequency, for these words, the positive correlation between score and embedding norm is less strong. The negative effect on the non-topic words may last longer.  When considering real world tasks, the structure of the tasks is complicated by many factors; it would be difficult to be conclusive in this regard.
> >
> > Besides, regarding the non-topic words like "the," "a," etc., it is common to remove them in the preprocessing step. Moreover, the experimental results (not included in the paper) show that removing such words does not obviously change our model's behavior.
> >
> > - *Lemma 1: while it is true that fixing q doesn't change the capacity of the model, it will definitely change its training dynamics (which is very much the theme of the paper as per the title). How important is it to fix q from this perspective?*
> >
> > 	**Response**: We agree that the training dynamic changes when we fix the query; however, it will not change much. As we have discussed in the previous response, regardless of the trainability of the query, the positive SEN relationship always exists as well as all the results based on it. Moreover, when the query is initialized with a large norm, the model's training dynamic is quite similar to the fixed query cases. We have made more discussions and provided more experimental results in Section C.1.
> >
> > - In 2. "[Hahn and Brunner] show theoretically that the self-attention blocks are severely limited in their computational capabilities when the input sequence is long, which implies that self-attention may not provide the interpretability that one expects.": can you clarify this sentence? Limitation in computational capabilities does not seem to entail limited interpretability in general (see linear models for instance).
> >
> > 	**Response**: “Limitation in computational capabilities” were choices of words in [Hahn and Brunner]. We think more precisely, this limitation refers to a non-identifiability of the attention model, where multiple weight configurations may give equally good end prediction. The non-uniqueness of the attention weights therefore makes it lack interpretability.
> >
> > - Regarding the problems related to the notations and typos.
> >
> > 	**Response**: We will make the corresponding changes in the revised version.

---

> > > ### Comment · AnonReviewer1 · 2020-11-20
> > > **Thank you for rebuttal + additional comments**
> > >
> > > First, I'd like to thank the authors for their in-depth response, and I appreciate that they took the time to address each of my concerns in such details.
> > >
> > > In my review, the main concerns were (1) validity of assumptions, (2) confusing writing/notation and (3) unclear takeaway. The rebuttal appropriately addressed (1), although I am looking forward to the revision to see how this is discussed in the paper itself. I cannot really say anything about any improvements on the writing (2) without seeing the revision, but I am confident that the authors can address most of the issues pointed out by myself and other reviewers. Regarding (3), unclear takeaway, after reading the authors' response as well as the other reviews, my concerns are somewhat assuaged (partly because the assumptions were addressed better), although I am still unsure how or if the results in this paper could be expanded to realistic attention models.
> > >
> > > Overall, I am willing to raise my score after seeing the revised paper (since a lot of my concerns have to do with writing ultimately)
> > >
> > > Some more specific remarks on the response:
> > >
> > > \> Having said this, in earlier attention-based models, indeed there exist abundant models (mostly for simpler tasks like text classification) in which query is chosen as a parameter independent of the input sentence, see, e.g., Wang et al, “Attention-based LSTM for Aspect-level Sentiment Classification”, ACL 2016. (Google Scholar Citations 830).
> > >
> > > This is true, but as far as I can tell these models are not cited in the paper, which currently cites Bahdanau et al., 2014 and Devlin et al., 2019 as references of "Attention-based neural networks". As a side note, Wang et al. (2016) does not use dot product attention but the more common (at the time) "MLP" style attention.
> > >
> > > \> Besides, regarding the non-topic words like "the," "a," etc., it is common to remove them in the preprocessing step.
> > >
> > > I don't think that this is true in the case of neural networks. This can certainly be a pre-processing step in topic modeling, or for simple tf-idf based linear model. But in general when using neural networks pre-processing includes tokenization/subword splitting but not stop-word removal (at least not in all the attention models cited in the paper). This specific point shouldn't be used as an argument in the paper.
> > >
> > > \> Hahn and Brunner
> > >
> > > My remark was more about how the current wording in *this* paper is confusing. With the expanded page limit in the final revision it is worth clarifying the citation in the text.
> > >
> > > Finally, here are additional remarks I had upon re-reading the paper:
> > >
> > > - Missing references with regards to "analyzing the behaviours of a well-trained attention-based models" which came out around the same time as Clark et al (2019)
> > >     - Voita, Elena, et al. "Analyzing Multi-Head Self-Attention: Specialized Heads Do the Heavy Lifting, the Rest Can Be Pruned." Proceedings of the 57th Annual Meeting of the Association for Computational Linguistics. 2019.
> > >     - Michel, Paul, Omer Levy, and Graham Neubig. "Are sixteen heads really better than one?." Advances in Neural Information Processing Systems. 2019.
> > > - In general the paper is very parsimonious with citations, specifically references to attention models for NLP (which is the core subject of the paper's analysis). On top of Bahdanau, Devlin and Vaswani, other important citations would be the Wang et al. (2016) paper mentioned above, but also for instance Luong et al. (2015) "Effective Approaches to Attention-based Neural Machine Translation" which, to the best of my knowledge, introduced the "MLP" style attention which was so ubiquitous in attention models pre-transformers (incidentally I don't think this type of attention is mentioned, and it should, even if it is not as prevalent anymore).
> > > - Minor comment on style: the phrase "Despite its great success established empirically, the working mechanism of attention has not been investigated at a sufficient theoretical depth to date" is paraphrased almost word for word in the introduction: "Although great successes of attention have been established empirically, its working mechanism has not been well understood"

---

> > > > ### Comment · AnonReviewer4 · 2020-11-20
> > > > **Citations**
> > > >
> > > > Important point about about the broad lack of citations to current analysis of attention in NLP. I definitely noticed Voita missing (I pointed it out in my review as well), but hadn't thought about the general trend. I might drop my score a bit, because the authors need to discuss current use and analysis of attention in NLP. I still don't think the assumptions are particularly overwhelming for a theory paper, but they should qualify their result by explaining how their assumptions diverge from current use of attention (particularly self-attention) in practice. If the interaction between attention weights and embeddings is relevant to the debate over whether attention is/is not/is not not explanation, it should be discussed.

---

> > > > > ### Author Response · Authors · 2020-11-21
> > > > > **Clarification**
> > > > >
> > > > > Thank you very much for the comments. Here are our responses.
> > > > >
> > > > > * *Important point about about the broad lack of citations to current analysis of attention in NLP. I definitely noticed Voita missing (I pointed it out in my review as well), but hadn't thought about the general trend. I might drop my score a bit, because the authors need to discuss current use and analysis of attention in NLP.*
> > > > >
> > > > > 	**Response**: We accept this criticism. This problem is in part due to the page limitation in the submitted. Nonetheless, this should not be an excuse for missing important works and we apologize for this sloppiness. In the revision,  we will make every effort to fully discuss the background literature and carefully position this work in the grand landscape of other works that analyze attention in NLP.
> > > > >
> > > > > 	On the other hand, we wish to note that we have been well aware of all works related to the analysis of attention, including the ones the reviewers bring up (it is our fault not discussing them). We argue that the existence of other analysis does not shadow the value of this work. Relative to those works (such as the debates on whether attention is an explanation and the different roles of multiple heads), this work provides a complementary perspective with a focus on the training dynamics. We do believe that this paper and our techniques therein may bring fresh insights and inspiration to the problem scope.
> > > > >
> > > > > 	We hope that the contribution of this paper outweighs our sloppiness in discussing the background literature, and that you trust we will make our best effort fixing this error.
> > > > >
> > > > >
> > > > > * *I still don't think the assumptions are particularly overwhelming for a theory paper, but they should qualify their result by explaining how their assumptions diverge from current use of attention (particularly self-attention) in practice.*
> > > > >
> > > > > 	**Response**: In the revision, we will better explain and justify the assumptions in this paper, and include some of the additional results which we provide during the rebuttal (Appendix C2, in the updated version). We will also make an effort to explain how the assumptions and our setup diverge from the current ways of using attention, and to what extent our setup is relevant (as we discussed in our response to initial review of AnonReviewer1).
> > > > >
> > > > > 	Regarding self-attention, we admit that we are not yet certain how our analysis may be exploited in the study for that setting. Studying that setting is of our greatest interest, but at present, our philosophy is to move one step at a time. We hope that this work will help to pave a way.
> > > > >
> > > > >
> > > > > * *If the interaction between attention weights and embeddings is relevant to the debate over whether attention is/is not/is not not explanation, it should be discussed.*
> > > > >
> > > > > 	**Response**: In the setting of the simple task in this paper, our results in fact affirm that attention IS an explanation. We agree that we should have explicitly stated this in the paper. Thank you for your suggestion.

---

> > > > ### Author Response · Authors · 2020-11-21
> > > > **Thank you very much for your detailed replies**
> > > >
> > > > We greatly appreciate your detailed and instructive replies. Here are our responses.
> > > >
> > > > * *In my review, the main concerns were (1) validity of assumptions, (2) confusing writing/notation and (3) unclear takeaway. The rebuttal appropriately addressed (1), although I am looking forward to the revision to see how this is discussed in the paper itself. I cannot really say anything about any improvements on the writing (2) without seeing the revision, but I am confident that the authors can address most of the issues pointed out by myself and other reviewers. Regarding (3), unclear takeaway, after reading the authors' response as well as the other reviews, my concerns are somewhat assuaged (partly because the assumptions were addressed better), although I am still unsure how or if the results in this paper could be expanded to realistic attention models.*
> > > >
> > > > 	**Response**: Thank you for your comments. We promise that we will make our best effort revising this paper, to streamline the ideas, clean the notations, improve the clarity and address all reviewers’ comments. It is also our hope that via a thorough revision, we can better disseminate these results to the community and inspire further studies that move forward the understanding of attention.
> > > >
> > > >
> > > > * *Overall, I am willing to raise my score after seeing the revised paper (since a lot of my concerns have to do with writing ultimately)*
> > > >
> > > > 	**Response**: Thank you very much!
> > > >
> > > > * *Some more specific remarks on the response: Having said this, in earlier attention-based models, indeed there exist abundant models (mostly for simpler tasks like text classification) in which query is chosen as a parameter independent of the input sentence, see, e.g., Wang et al, “Attention-based LSTM for Aspect-level Sentiment Classification”, ACL 2016. (Google Scholar Citations 830).*
> > > >
> > > > 	*This is true, but as far as I can tell these models are not cited in the paper, which currently cites Bahdanau et al., 2014 and Devlin et al., 2019 as references of "Attention-based neural networks". As a side note, Wang et al. (2016) does not use dot product attention but the more common (at the time) "MLP" style attention.*
> > > >
> > > > 	**Response**: We agree that our citations of related works were thin and sloppy. Indeed, the papers you point to (here and later) should be cited to display the full spectrum of attention based models. We will follow your advice and include all references brought up by all reviewers.
> > > >
> > > > 	We agree with your comments on Wang et al (2016). The attention mechanism used there is indeed more complex than the model in this paper, although it shares some similarity with our model in that in both cases, the query is independent of the input. The objective of this paper is to develop a concrete understanding of attention, starting from the simplest case. We believe that this is a necessary step towards understanding more complex attention models, such as Wang et al (2016) and transformers.
> > > >
> > > >
> > > > * *Besides, regarding the non-topic words like "the," "a," etc., it is common to remove them in the preprocessing step.*
> > > >
> > > > 	*I don't think that this is true in the case of neural networks. This can certainly be a pre-processing step in topic modeling, or for simple tf-idf based linear model. But in general when using neural networks pre-processing includes tokenization/subword splitting but not stop-word removal (at least not in all the attention models cited in the paper). This specific point shouldn't be used as an argument in the paper.*
> > > >
> > > > 	**Response**: Thank you for pointing this out. We agree that in neural models it is unnecessary to remove the stop words. We accept this correction.
> > > >
> > > > *	*Hahn and Brunner*
> > > >
> > > > 	*My remark was more about how the current wording in this paper is confusing. With the expanded page limit in the final revision it is worth clarifying the citation in the text.*
> > > >
> > > > 	**Response**: We agree that our wording there hasn’t been crisp. We will certainly revise it!

---

> > > > > ### Author Response · Authors · 2020-11-21
> > > > > **Thank you very much for your detailed replies (cont.)**
> > > > >
> > > > > * *Finally, here are additional remarks I had upon re-reading the paper:*
> > > > > 	*	*Missing references with regards to "analyzing the behaviours of a well-trained attention-based models" which came out around the same time as Clark et al (2019)*
> > > > > 		* *Voita, Elena, et al. "Analyzing Multi-Head Self-Attention: Specialized Heads Do the Heavy Lifting, the Rest Can Be Pruned." Proceedings of the 57th Annual Meeting of the Association for Computational Linguistics. 2019.*
> > > > > 		* *Michel, Paul, Omer Levy, and Graham Neubig. "Are sixteen heads really better than one?." Advances in Neural Information Processing Systems. 2019.*
> > > > >
> > > > > 	*In general the paper is very parsimonious with citations, specifically references to attention models for NLP (which is the core subject of the paper's analysis). On top of Bahdanau, Devlin and Vaswani, other important citations would be the Wang et al. (2016) paper mentioned above, but also for instance Luong et al. (2015) "Effective Approaches to Attention-based Neural Machine Translation" which, to the best of my knowledge, introduced the "MLP" style attention which was so ubiquitous in attention models pre-transformers (incidentally I don't think this type of attention is mentioned, and it should, even if it is not as prevalent anymore).*
> > > > >
> > > > > 	**Response**: We apologize that our citations of the existing works are not adequate. We hope that you have faith in our ability and desire to fix this in the revision.
> > > > >
> > > > > * *Minor comment on style: the phrase "Despite its great success established empirically, the working mechanism of attention has not been investigated at a sufficient theoretical depth to date" is paraphrased almost word for word in the introduction: "Although great successes of attention have been established empirically, its working mechanism has not been well understood"*
> > > > >
> > > > > 	**Response**: We will revise this.
> > > > >
> > > > >
> > > > > **Summary:** Finally, we would like to express our great appreciation for your effort in providing a careful review and many valuable comments. We are not able to implement a sufficiently satisfactory revision, addressing all your comments,  during the short window of the rebuttal period. But we hope that you trust we will make every effort in that direction. If there is anything else that you believe we must address immediately during the rebuttal period so as to bring the paper to an acceptable level in your standard, please advise. We will try our best to address that before the rebuttal is closed.

---

### Official Review · AnonReviewer2 · 2020-10-29
**Review for On The Dynamics of Training Attention Models**

**Rating:** 7
**Confidence:** 2

**Review:**

(Summary)

The paper investigates the dynamics of attention mechanism by configurating a controlled experiment on a simple topic classification task and training via gradient descent. Each random sentence in the training data is synthesized to include only one topic word among many. Then the authors try to find an intrinsic mechanism that triggers the attention model to discover the topic word and accelerates training via mutual promotion. They further experiment the evolution of models during optimization when no clear distinction between topic and non-topic words exist like in real data.


(Originality and Contribution)

The paper proposes an artificial topic classification task and shows a positive score-and-embedding-norm relationship for the topic words to which the model must attend to. The authors also show that attention mechanism is highly helpful when the classifier has only limited capacity. They also demonstrate mutual promotion effect that leads a faster dropping of training loss than the fixed score and fixed embeddings. This discovery sounds to be original and relevant contribution to the field.


(Strength and Weakness)
-	Strength: Design and run novel controlled experiment. Extensive analysis.
-	Weakness: Too much notational overloading. Writing quality.


(Concerns, Questions, and Suggestions)

1) It is unclear why $M >> N$ implies that a topic word appears more frequently in the sentences than a non-topic word. Section 3 describes that each sentence consists of only one topic word, then combining with $m$ non-topic words drawn uniformly at random. When the total number of topics $N$ is much smaller than the size of non-topic word dictionary $M$, what increases frequency of topic word?

2) Overall simplifying some notations and avoiding notational overloading would greatly increase the readability of the paper.

3) To reduce confusion, it would be great to change the iterator of summation for the partition function Z into $\sum_{w’ \in S_k}$ rather than using the same $w$.

4) In Lemma 1, assume $q \neq 0$ -> $q \neq \vec{0}$.

---

> ### Author Response · Authors · 2020-11-13
> **Clarification**
>
> Thank you for your reviews! We believe you have precisely captured the main concepts of our paper. We have carefully gone through them, and here are our responses and explanations.
>
> - It is unclear why $M >> N$ implies that a topic word appears more frequently in the sentences than a non-topic word. Section 3 describes that each sentence consists of only one topic word, then combining with  $m$ non-topic words drawn uniformly at random. When the total number of topics $N$ is much smaller than the size of non-topic word dictionary $M$, what increases frequency of topic word?
>
> **Response**: We will use an example to clarify this point. For simplicity, assume we have $N$ topics and each topic only has one topic word (so we have $N$ topic words in total). Consider the sentence generating process introduced in the paper. For a given topic word, the probability that the topic word appears in a sentence is $1/N$, while the probability of a given non-topic word appearing in a sentence of length $m$ is $1-((M-1)/M)^m$. As a result, for a fixed $m$ and $N$,  if $M >> N$, $1-((M-1)/M)^m$ approaches to zero. Thus a topic word appears more frequently in a sentence than a non-topic word.
>
>
> - Other problems regarding notations and writing quality.
>
> **Response**: We will simplify our notations and make the manuscript flow better in the revised version.

---

### Official Review · AnonReviewer3 · 2020-11-02
**Not a sizeable contribution**

**Rating:** 4
**Confidence:** 3

**Review:**

This paper studies the dynamics of attention in a task of simplified topic modeling, over the course of training for a specific model, where the context vector is the sum over words in a sentence of their embedding weighted by the exponential of the dot-product their key embedding with a global query vector, normalized. Due to the simplification of the topic modeling problem (two null-intersect sets of words: topic vs. non-topic), they consider the embeddings of the non-topic words to be fixed over the course of training for their theoretical analysis. The applicability of the theoretical result is close to zero, and a somewhat known property (e.g. in word2vec, Mikolov et al. 2013). The experimental results include two parts. One on a tiny synthetic dataset that matches the simplified topic modeling problem and serves as illustration. The other is on SST2 and SST5 (movie comments and ratings, sentiment analysis), where the results are poor (obviously, as the model is simple), e.g. yielding 79.59% on SST while the SOTA is 97.4, and BERT base is at 91.2. The analysis is interesting, but does not lead to new insights.

For a simple analysis, the paper is at times hard to follow, and could benefit from more structure (presenting "what" before "how") and better notation (e.g. \nu, v, $v$ all attached to (forms of the) the context vector).

Overall, the contribution does not seem sufficient enough for inclusion at ICLR. The paper could be a good fit for a workshop on topic modeling or attention-based models.

---

> ### Author Response · Authors · 2020-11-13
> **Perhaps not a matter of contribution but a matter of taste**
>
> Thank you for your review! We have carefully read them and here are our responses and explanations.
>
> - *The applicability of the theoretical result is close to zero, and a somewhat known property (e.g. in word2vec, Mikolov et al. 2013). The experimental results include two parts. One on a tiny synthetic dataset that matches the simplified topic modeling problem and serves as illustration. The other is on SST2 and SST5 (movie comments and ratings, sentiment analysis), where the results are poor (obviously, as the model is simple), e.g. yielding 79.59% on SST while the SOTA is 97.4, and BERT base is at 91.2. The analysis is interesting, but does not lead to new insights.*
>
> **Response**: We wish to note that the purpose of this work is not to produce SOTA results for specific NLP tasks. Rather our sole interest is to understand how attention works. Despite the nearly universal applicability of the attention module in neural net based language models, its working mechanism is poorly understood to date. In this work, our hope is to characterize the dynamics of training attention models, using a simplified task. Although research at such a fundamental level may appear somewhat distant from immediate applications, it contributes to developing concrete insights and deep understanding in neural networks, and potentially has a long term impact.   Without such understanding and insights, neural networks or deep learning may remain as black boxes, and practitioners can only rely on heuristics in model design and alchemy-like hyper-parameter tuning in model training. For this reason, we can not agree that the applicability of this research is zero. It is unfortunate that our disagreement on this matter perhaps has merely reflected our distinct research taste from yours.
>
> - *For a simple analysis, the paper is at times hard to follow, and could benefit from more structure (presenting "what" before "how") and better notation.*
>
> **Response**: We will make an effort to simplify our notations and make the manuscript flow better in the revised version.

---

### Decision · Program_Chairs · 2021-01-07
**Final Decision**

**Decision:**

Accept (Poster)

**Comment:**

This paper investigates the training dynamics of simple neural attention
mechanisms, in a controlled setting with clear (but rather strict) assumptions. Some reviewers
expressed caution about the applicability of the assumptions in practice,
but nevertheless there is agreement that the results deepen our understanding and enrich
our toolkit for reasoning about attention.
In support of this, in the discussion period, it was emphasized that the work uses different techniques than
most current work in this direction. I am therefore confident that the paper will be useful, and recommend acceptance.

I strongly encourage the authors to improve the clarity of the work and thorough
citation, as suggested by the reviewers.